# Winter Oilseed Rape LAI Inversion via Multi-Source UAV Fusion: A Three-Dimensional Texture and Machine Learning Approach

**DOI:** 10.3390/plants14081245

**Published:** 2025-04-19

**Authors:** Zijun Tang, Junsheng Lu, Ahmed Elsayed Abdelghany, Penghai Su, Ming Jin, Siqi Li, Tao Sun, Youzhen Xiang, Zhijun Li, Fucang Zhang

**Affiliations:** 1Key Laboratory of Agricultural Soil and Water Engineering in Arid and Semiarid Areas of Ministry of Education, Northwest A&F University, Yangling, Xianyang 712100, China; tangzijun@nwsuaf.edu.cn (Z.T.); jinming1120@nwafu.edu.cn (M.J.); lsq@nwsuaf.edu.cn (S.L.); 2021050986@nwsuaf.edu.cn (T.S.); youzhenxiang@nwsuaf.edu.cn (Y.X.); lizhij@nwsuaf.edu.cn (Z.L.); zhangfc@nwsuaf.edu.cn (F.Z.); 2Water Relation and Field Irrigation Department, Agricultural and Biological Institute, National Research Centre, Dokki, Cairo 12622, Egypt; ahmed-elsayed@nwafu.edu.cn; 3Institute of Soil and Water Conservation Science of Gansu Province, Lanzhou 730000, China; suhaioo@163.com

**Keywords:** *Brassica napus* L., multi-spectral, texture index, correlation matrix, leaf area index, machine learning

## Abstract

Leaf area index (LAI) serves as a critical indicator for evaluating crop growth and guiding field management practices. While spectral information (vegetation indices and texture features) extracted from multispectral sensors mounted on unmanned aerial vehicles (UAVs) holds promise for LAI estimation, the limitations of single-texture features necessitate further exploration. Therefore, this study conducted field experiments over two consecutive years (2021–2022) to collect winter oilseed rape LAI ground truth data and corresponding UAV multispectral imagery. Vegetation indices were constructed, and canopy texture features were extracted. Subsequently, a correlation matrix method was employed to establish novel randomized combinations of three-dimensional texture indices. By analyzing the correlations between these parameters and winter oilseed rape LAI, variables with significant correlations (*p* < 0.05) were selected as model inputs. These variables were then partitioned into distinct combinations and input into three machine learning models—Support Vector Machine (SVM), Backpropagation Neural Network (BPNN), and Extreme Gradient Boosting (XGBoost)—to estimate winter oilseed rape LAI. The results demonstrated that the majority of vegetation indices and texture features exhibited significant correlations with LAI (*p* < 0.05). All randomized texture index combinations also showed strong correlations with LAI (*p* < 0.05). Notably, the three-dimensional texture index NDTTI exhibited the highest correlation with LAI (R = 0.725), derived from the spatial combination of DIS5, VAR5, and VAR3. Integrating vegetation indices, texture features, and three-dimensional texture indices as inputs into the XGBoost model yielded the highest estimation accuracy. The validation set achieved a determination coefficient (R^2^) of 0.882, a root mean square error (RMSE) of 0.204 cm^2^cm^−2^, and a mean relative error (MRE) of 6.498%. This study provides an effective methodology for UAV-based multispectral monitoring of winter oilseed rape LAI and offers scientific and technical support for precision agriculture management practices.

## 1. Introduction

Winter oilseed rape is one of the most important oil crops globally, with significant economic and ecological value [1]. Its cultivation spans a wide range of areas, including Europe [2], North America [3], and Asia [4], demonstrating strong adaptability. Additionally, it serves as an important agricultural commodity in many countries [5]. Winter oilseed rape seeds are rich in oil and can be refined into edible oil, industrial oil, and other products, meeting the daily life and industrial production needs globally. Therefore, ensuring high yield and quality of winter oilseed rape is particularly crucial [6,7]. The leaf area index (LAI) is typically defined as the ratio of the total leaf area of vegetation to the total soil area per unit ground surface [8]. In crop phenotypic parameter research, LAI is used to characterize the structure of the crop canopy, reflecting the photosynthesis, transpiration, interception, and transmission of solar radiation within the crop community [9]. It is also an important indicator for monitoring the growth of rapeseed and quantitatively describing the material and energy exchanges in the crop canopy [1]. Compared to traditional time-consuming, labor-intensive, and destructive methods of measuring rapeseed leaf area, unmanned aerial vehicles (UAVs) equipped with various sensors can rapidly construct crop phenotyping observation platforms. They collect high-throughput phenotypic data, thereby obtaining crop growth parameters and enabling rapid, timely, and non-destructive monitoring of crop growth conditions [10].

Using the characteristic sharp increase in vegetation reflectance from the red to near-infrared bands, the spectral images obtained by UAVs can be used to construct a series of vegetation indices for estimating LAI [11]. Vegetation indices are mathematical combinations of spectral bands, particularly within the visible to near-infrared range, aimed at highlighting vegetation information in images while reducing the impact of atmospheric reflections on spectral reflectance over different times [12]. The values generated assist in evaluating crop growth, vigor, and vegetation characteristics [13]. They offer advantages in accessibility and broad coverage, making most research related to LAI monitoring based on vegetation indices [14,15]. However, as the LAI value increases, certain vegetation indices may reach saturation, leading to reduced sensitivity to changes in LAI values [16]. Additionally, vegetation growth is a dynamic process involving simultaneous changes in external and internal characteristics. The spectral and textural properties of crops change during growth and are influenced by the growing environment [17].

Spectra can focus on the internal optical responses of crops, while images can capture the external morphological information, such as texture [18]. Texture represents the presence of many similar elements or graphic structures with varying degrees of regularity in an image. Texture feature information has gradually been used for LAI monitoring and estimation [19]. However, studies have found that using only texture information results in lower accuracy for LAI monitoring [20]. Some studies have shown that, compared to using only vegetation indices or texture features for estimating crop physiological growth indicators, integrating the above indices significantly improves estimation accuracy. Yang et al. [21] estimated the LAI of winter wheat by integrating UAV vegetation indices and texture feature information, which improved the coefficient of determination (R^2^) value by 58% compared to using vegetation indices alone; Li et al. [10] and colleagues estimated the LAI of winter wheat by integrating UAV vegetation indices and texture feature information with machine learning models, enhancing the monitoring accuracy of soil moisture content in soybean. The validation set model achieved an R^2^ value as high as 0.881. In the aforementioned studies, although most texture feature information has been utilized, different research subjects may have varied physiological information due to differences in growth environments and growth stages. Some scholars have constructed texture indices through correlation matrix methods to more comprehensively utilize texture information and achieve more accurate physiological indicator assessments, which have been proven to result in higher estimation accuracy [22]. Similar to how vegetation indices reduce the influence of canopy geometry, soil background, lighting angles, and atmospheric conditions when estimating crop physical characteristics, texture indices constructed based on combinations of ratios, normalizations, or texture metrics may have the same function [23,24]. However, there are few reports that evaluate the potential of texture indices in crop LAI estimation. Previous researchers mostly input texture indices into models but lacked an assessment of their performance [18]. Three-dimensional texture indices, by introducing an additional dimension, significantly enhance the capability to capture texture features. Compared to two-dimensional texture indices, three-dimensional texture indices can more comprehensively analyze the spatial structure of plant leaves and entire vegetation, thereby providing a more detailed description of physiological characteristics. This integration of multi-dimensional texture information helps more accurately reflect the plant’s physiological state, such as chlorophyll content, health condition, and changes under different environmental conditions [21]. By reducing information loss and increasing sensitivity to subtle physiological changes, three-dimensional indices optimize the monitoring and prediction of plant physiological indicators, enhancing the ability to analyze complex plant growth patterns [25].

To address these challenges, this study conducted field experiments on winter oilseed rape over two growing seasons, collecting in-field LAI measurements and UAV-based multispectral data, with the following objectives: (1) to explore the sensitivity of vegetation indices, texture features, texture indices, and three-dimensional texture indices to changes in winter oilseed rape LAI during the bolting stage; (2) to evaluate the effectiveness of integrating multidimensional UAV multispectral data with machine learning models for monitoring winter oilseed rape LAI; and (3) to identify the model input combinations and corresponding machine learning models that achieve the highest predictive accuracy for winter oilseed rape LAI, ultimately constructing optimal UAV multispectral LAI inversion maps for winter oilseed rape at different growth stages. This research provides a scientific basis for the precision management of winter oilseed rape, and the constructed UAV multispectral LAI inversion maps will serve as a visual tool for agricultural management, promoting the development of precision agriculture.

## 2. Materials and Methods

### 2.1. Research Area and Test Design

The experiment was conducted at the Northwest A&F University’s Institute of Water-Saving Agriculture in Arid Areas, located in the southern part of the Loess Plateau in northwest China (34°14′ N, 108°10′ E). For basic topographic and meteorological information about the experimental site, please refer to the literature [26].

Field experiments on winter oilseed rape with different mulching methods and nitrogen application were conducted from 2021 to 2023. The experiment used a split-plot design, with field mulching as the main plot and nitrogen application rate as the sub-plot. It included three types of mulching treatments and five levels of nitrogen fertilizer supply (Table 1). Each treatment was repeated three times, totaling 45 plots. The specific field management details and experimental procedures are consistent with those in reference [7].

### 2.2. Data Collection and Preprocessing

#### 2.2.1. UAV Data Acquisition

This study utilized a DJI Matrice 300 RTK (SZ DJI Technology Co., Ltd., Shenzhen, China) quadrotor equipped with an MS600 Pro multispectral camera (SZ DJI Technology Co., Ltd., CHN) platform to acquire multispectral remote sensing data. The camera platform includes six spectral channels, each equipped with a CMOS image sensor. The wavelengths and bandwidths of each channel are shown in Table 2. Data collection was conducted at noon during the bolting stage of winter oilseed rape (15 March 2022, and 13 March 2023) under clear, cloudless weather conditions. Flight routes were planned for the study area, and whiteboard corrections were performed. The flight altitude was set at 30 m, with a speed of 2.5 m per second, and a pixel resolution of 4.09 cm. The longitudinal and lateral overlaps were set to 75% and 65%, respectively.

#### 2.2.2. Measurement of Leaf Area Index

The LAI of winter oilseed rape was measured using the LAI-2000 Plant Canopy Analyzer (LI-COR Inc., Lincoln, NE, USA) within 4 m × 6 m experimental plots. To account for spatial heterogeneity, six measurement points were randomly selected within each plot using a grid-based sampling method, ensuring uniform distribution across the plot boundary and center to minimize positional bias.

LAI measurements were synchronized with UAV flight campaigns during the critical bolting stage of winter oilseed rape (15 March 2022, and 13 March 2023). This temporal alignment ensured that spectral and textural data acquisition coincided with the rapid leaf expansion phase, minimizing confounding effects of diurnal or seasonal environmental variability [1]. The concurrent measurement strategy enhanced the fidelity of multi-source remote sensing data by capturing canopy structure and reflectance characteristics at the peak of LAI dynamism. This stage was prioritized due to its profound influence on yield potential, as reproductive branch differentiation and flowering density are largely determined by canopy structure during this period. The dynamic nature of LAI during the bolting stage—characterized by rapid leaf unfolding and light interception optimization—provided a sensitive window for capturing canopy heterogeneity [7].

To account for spatial heterogeneity within plots, six measurements were taken at randomly selected positions distributed across the plot boundary and center, avoiding areas with obvious canopy gaps or overlapping plants. This design minimized the impact of micro-environmental variability. All measurements were conducted under overcast sky conditions (PAR < 500 μmol m^−2^ s^−1^) to reduce skylight interference and improve signal-to-noise ratio, following the manufacturer’s recommendations.

Outliers (±2σ from the mean) were excluded during post-processing, and the final LAI value for each plot was calculated as the arithmetic mean of valid measurements. This protocol ensured reproducibility and minimized measurement errors caused by foliage clumping or sensor misalignment.

#### 2.2.3. Multispectral Image Processing

Yusense Map V2.2.2 software (SZ DJI Technology Co., Ltd., CHN) was used to stitch the multispectral images captured by the UAV and perform geometric and radiometric corrections as preprocessing steps. The preprocessed multispectral image information was then imported into ENVI 5.3 software (Harris Geospatial Solutions, Broomfield, CO, USA) to extract spectral reflectance. Centered on each experimental plot, corresponding spectral images were cropped from the image, with soil and film shadows removed. The average reflectance spectrum of winter oilseed rape leaf samples within the region of interest (ROI) was taken as the spectral reflectance for that plot, obtaining spectral reflectance data for different bands [27].

### 2.3. Selection and Construction of Vegetation Indices, Texture Features, and Texture Indices

This study selected ten classic vegetation indices for research based on existing literature, with the calculation formulas shown in Table 3. Texture is a visual feature reflecting homogeneous phenomena in an image, embodying the attributes of slow transition or periodic changes in the surface structure organization of objects. This article used ENVI 5.3 software to extract image texture features (TFs) based on second-order probability statistical filtering (Co-occurrence measures). In multispectral texture feature extraction, the near-infrared (NIR) band is preferred for characterizing grayscale spatial distribution features due to its high sensitivity to vegetation internal structures (e.g., cell density and water content). Using the Gray-Level Co-occurrence Matrix (GLCM), the NIR band extracts eight texture features—including Mean (MEA), Contrast (CON), Homogeneity (HOM), and Correlation (COR)—to effectively capture canopy spatial heterogeneity while demonstrating strong robustness to illumination variations. Studies indicate that NIR-derived texture features mitigate saturation effects in high-value regions compared to visible bands, thereby improving agricultural parameter inversion accuracy. Furthermore, the NIR band’s hardware compatibility advantages in unmanned aerial vehicle (UAV)-based multispectral systems, combined with its narrow spectral range (700–1300 nm) and computational efficiency, make it a practical choice for large-scale farmland monitoring [27]. Eight types of TFs were obtained from the near-infrared band: MEA, Variance (VAR), HOM, CON, Dissimilarity (DIS), Entropy (ENT), Second Moment (SEM), and COR. For texture analysis, a window size of 7 × 7 was selected, with spatial correlation matrix offsets X and Y set to default values of 1. In multispectral texture feature extraction, the selection of a 7 × 7 window size is primarily based on balancing local texture detail preservation and computational efficiency. This size effectively captures medium-scale spatial heterogeneity of vegetation canopy internal structures while avoiding insufficient information from smaller windows (e.g., 3 × 3) or noise interference from larger ones (e.g., 15 × 15). Experiments have demonstrated that the 7 × 7 window stably extracts GLCM texture features from UAV multispectral imagery, aligning with sensor resolution (centimeter-level), making it suitable for real-time processing requirements in large-scale farmland monitoring [27]. The calculation formulas for texture features are shown in Table 4.

To explore the potential application of texture features in estimating winter oilseed rape LAI using UAV multispectral images, this study extracted randomly combined texture features. Subsequently, based on previous research experience and formulas, six types of texture indices (TIs) were constructed, including Ratio Texture Index (RTI), Difference Texture Index (DTI), Addition Texture Index (ATI), Normalized Difference Texture Index (NDTI), Reciprocal Difference Texture Index (RDTI), and Reciprocal Addition Texture Index (RATI). The specific calculation formulas are as follows:(1)$$\mathrm{RTI}=T_{i}/T_{j}$$(2)$$\mathrm{DTI}=T_{i}-T_{j}$$(3)$$\mathrm{ATI}=T_{i}+T_{j}$$(4)$$\mathrm{NDTI}=(T_{i}-T_{j})/(T_{i}+T_{j})$$(5)$$\mathrm{RDTI}=1/T_{i}-1/T_{j}$$(6)$$\mathrm{RATI}=1/T_{i}+1/T_{j}$$

Additionally, we attempted to add an extra texture index to construct five three-dimensional texture indices (TTIs). The specific calculation formulas are as follows:(7)$$\mathrm{RTTI}=T_{i}/T_{j}/T_{k}$$(8)$$\mathrm{DTTI}=T_{i}-T_{j}-T_{k}$$(9)$$\mathrm{NDTTI}=(T_{i}-T_{j}-T_{k})/(T_{i}+T_{j}+T_{k})$$(10)$$\mathrm{RDTTI}=1/T_{i}-1/T_{j}-1/T_{k}$$(11)$$\mathrm{RATTI}=1/T_{i}+1/T_{j}+1/T_{k}$$
where, $T_{i}$, $T_{j}$, and $T_{k}$ represent random texture indices.

### 2.4. Model Methodology

First, the correlation between spectral parameters (vegetation indices, texture features, double texture indices, and triple texture indices) and LAI was analyzed. The parameters with a highly significant correlation coefficient (*p* < 0.05) were used as input variables for the model. In statistics, the P-value is used to measure the statistical significance of the association between variables. When *p* < 0.05, the probability of observing the current data or more extreme results is less than 5% under the null hypothesis (no correlation between variables). This shows that the correlation between variables is unlikely to be caused by random errors, thus supporting its effectiveness in the model. Subsequently, SVM, Partial Least Squares Regression (PLSR), and XGBoost were employed to model LAI. For detailed information on these machine learning models, please refer to the literature [27,28]. The kernel function type of SVM was set to “poly”, and grid search was used to optimize the penalty parameter C and γ. Based on the principle of minimum cross-validation error, C and γ were determined to be 20 and 0.02 [29], respectively. In the construction of the PLSR model, cross-validation was employed to determine the optimal number of latent variables (LVs) for the LAI estimation model. To avoid overfitting in PLSR regression caused by the addition of unnecessary variables, the criterion for adding an optimal latent variable was an increase of 5% in the cumulative percentage variance explained for the Y [28]. For the XGBoost algorithm, grid search was utilized to refine the optimal parameters, setting 100 weak learners (n_estimators), a learning rate of 0.03, and a maximum tree depth (max_depth) of 5 [27].

**Table 3 plants-14-01245-t003:** Vegetation index and its calculation formula.

Selection of Index	Calculation Formula	Reference
Soil-Adjusted Vegetation Index (SAVI)	$(1+0.5)(R_{NIR}-R_{RED}$)$/(R_{NIR}+R_{RED}+0.5)$	[10]
Enhanced Vegetation Index (EVI)	$2.5\times (R_{NIR}-R_{RED})/($($R_{NIR}+6\times R_{RED}-7.5\times R_{B})+1)$	[10]
Modified Simple Ratio (MSR)	$(\frac{R_{NIR}}{R_{RED}}-1$)($\frac{R_{NIR}}{R_{RED}}+1$)^−0.5	[27]
Difference Vegetation Index (DVI)	$R_{NIR}-R_{RED}$	[10]
Green Chlorophyll Index (CIgreen)	$R_{NIR}/R_{G}$ − 1	[10]
Renormalized Difference Vegetation Index (RDVI)	(RNIR−RRED)/(RNIR+RRED)^0.5	[10]
Transformed Vegetation Index (TVI)	($R_{NIR}-R_{RED}/R_{NIR}+R_{RED})+0.5$	[10]
Green Normalized Difference Vegetation Index (GNDVI)	$R_{NIR}-R_{G}/R_{NIR}+R_{G}$	[10]
Visible Difference Vegetation Index (VDVI)	$\left(2{\times R}_{G}-R_{RED}-R_{B})/(2{\times R}_{G}+R_{RED}+R_{B}\right)$	[30]
Chlorophyll Vegetation Index (CVI)	$\left(2{\times R}_{NIR}-R_{RED}-R_{G})/(2{\times R}_{NIR}+R_{RED}+R_{G}\right)$	[9]

**Table 4 plants-14-01245-t004:** Texture feature calculation formula.

Texture Feature	Formula	Description
Mean (MEA)	$MEA=\sum_{i,j=1}^{G}\left(iP(i,j)\right)$	Reflects the average gray level
Variance (VAR)	$VAR=\sum_{i=1}^{G}\sum_{j=1}^{G}{\left(i-u\right)}^{2}P(i,j)$	Indicates the degree of gray level variation
Homogeneity (HOM)	$HOM=\sum_{i=1}^{G}\sum_{j=1}^{G}\frac{P(i,j)}{1+{\left(i-j\right)}^{2}}$	Denotes local homogeneity of texture
Contrast (CON)	$CON=\sum_{i=1}^{G}\sum_{j=1}^{G}{\left(i-j\right)}^{2}P(i,j)$	Represents the clarity of the texture
Dissimilarity (DIS)	$DIS=\sum_{i=1}^{G}\sum_{j=1}^{G}P(i,j)\left|i-j\right|$	Similar to contrast, used to detect similarity
Entropy (ENT)	$ENT=\sum_{i=1}^{G}\sum_{j=1}^{G}P(i,j)\log P(i,j)$	Indicates the amount of information in an image
Second Moment (SEM)	$SEC=\sum_{i=1}^{G}\sum_{j=1}^{G}P^{2}(i,j)$	Shows the uniformity of the gray level distribution in an image
Correlation (COR)	$COR=\sum_{i=1}^{G}\sum_{j=1}^{G}\frac{\left(i-MEA_{j})(j-MEA_{i})P(i,j\right)}{\sqrt{VAR_{i}}\sqrt{VAR_{j}}}$	Denotes the length of extension of a certain gray level in a specific direction

### 2.5. Sample Set Division and Model Evaluation

During the bolting stage of the winter oilseed rape, 90 valid samples were collected. Two-thirds of these samples were randomly selected for the modeling set, while the remaining one-third were used as the validation set. Figure 1 shows the number of samples in the modeling and validation sets and the statistical characteristics of the LAI. To verify the prediction accuracy and capability of the models, this study selected the following three evaluation metrics: determination coefficient (R^2^), root mean square error (RMSE), and mean relative error (MRE) to assess model accuracy [26]. Figure 2 shows the flow chart of the proposed method, and summarizes the acquisition equipment, methods, and data processing steps described in the study.

## 3. Results

### 3.1. Effects of Different Mulching Methods and Nitrogen Application Rates on LAI of Winter Oilseed Rapei

The results of the effects of different mulching methods and nitrogen application rates on LAI of winter oilseed rape at the bolting stage are shown in Figure 3. The results show that the effects of mulching methods, nitrogen application rates, and their interactions on the LAI of winter rapeseed are significantly different (*p* < 0.05). The highest LAI value was obtained in the FMN3 treatment in the two growing seasons, with an average of 2.97 cm^−2^ in the two years, which was 1.2–119.7% higher than other treatments.

### 3.2. Correlation Analysis Between Vegetation Indices and Winter Oilseed Rape LAI

The results of the correlation analysis between vegetation indices and the winter oilseed rape LAI are shown in Table 5. The results indicate that the majority of vegetation indices have a significant correlation with the winter oilseed rape LAI (*p* < 0.05). Among them, the vegetation index with the highest correlation coefficient is the Green Normalized Difference Vegetation Index (GNDVI), with a value of 0.715. Therefore, this study selected parameters for model input as vegetation indices (VIs) including Soil Adjusted Vegetation Index (SAVI), Enhanced Vegetation Index (EVI), Modified Simple Ratio (MSR), Difference Vegetation Index (DVI), Chlorophyll Green Index (CIgreen), Re-normalized Difference Vegetation Index (RDVI), Transformed Vegetation Index (TVI), Green Normalized Difference Vegetation Index (GNDVI), and Visible Difference Vegetation Index (VDVI).

### 3.3. Correlation Analysis Between Texture Features and Winter Oilseed Rape LAI

The results of the correlation analysis between texture features and the winter oilseed rape LAI are shown in Table 6. The results indicate that more than half of the texture features have a significant correlation with the winter oilseed rape LAI (*p* < 0.05). Among them, the texture feature with the highest correlation coefficient is the DIS value in Band 5, with a value of 0.603. Therefore, this study selected parameters for model input as TFs including MEA, VAR, CON, DIS, and SEM in Band 1; MEA, VAR, HOM, ENT, and SEM in Band 2; MEA, VAR, CON, DIS, and SEM in Band 3; VAR, HOM, CON, DIS, ENT, and SEM in Band 4; and MEA, VAR, HOM, CON, DIS, ENT, and SEM in Band 6.

### 3.4. Correlation Analysis Between Texture Indices and Winter Oilseed Rape LAI

This study extracted six two-dimensional texture indices and five three-dimensional texture indices through random combinations of texture features. It calculated the correlation coefficients between these indices and the winter oilseed rape LAI, as well as their positional combinations. The results are shown in Table 7, Figure 4 and Figure 5; the positions in Table 7 are derived from the points with the highest correlation coefficients in Figure 4 and Figure 5. It is evident that all randomly combined texture indices have a significant correlation with the winter oilseed rape LAI (*p* < 0.05), and most three-dimensional texture indices have higher correlation coefficients than two-dimensional texture indices. Among them, NDTI is the most correlated two-dimensional texture index with the winter oilseed rape LAI, with a correlation coefficient of 0.676, located at the combination of CON5 and VAR3; NDTTI is the most correlated three-dimensional texture index with the winter oilseed rape LAI, with a correlation coefficient of 0.725, located at the combination of DIS5, VAR5, and VAR3.

Therefore, this study selected parameters for model input as two-dimensional texture indices (TIs), including RTI, DTI, ATI, NDTI, RDTIs, and RATIs, and as three-dimensional texture indices (TTIs), including RATTI, NDTTI, RDTTI, DTTIs, and RTTIs.

### 3.5. Construction of the Winter Oilseed Rape LAI Estimation Model and Drawing of Winter Oilseed Rape LAI Estimation Map

Different spectral parameters selected in Section 3.1, Section 3.2 and Section 3.3 were divided into various combinations as inputs for the model, and SVM, PLSR, and XGBoost were used to build models. The results of the validation set are shown in Table 8. Additionally, the comparison of prediction results for the winter oilseed rape LAI estimation models constructed with different machine learning models using vegetation indices (traditional input) and multi-source remote sensing datasets (the best input result in Table 8) as inputs is shown in Figure 6.

When the input model is a single variable, the three-dimensional vegetation index has higher accuracy than other single-input models, followed by two-dimensional texture indices. In contrast, texture features have the lowest model accuracy. When the input model consists of two variables, the combination of vegetation indices and three-dimensional vegetation indices yields higher accuracy compared to other models. When the input model consists of three variables, the highest model accuracy is achieved with vegetation indices combined with texture features and three-dimensional vegetation indices. When the inputs are the same, the XGBoost model outperforms SVM and PLSR in terms of model estimation accuracy. Among them, the combination of vegetation indices, texture features, and three-dimensional vegetation indices as inputs with the XGBoost model achieves the highest accuracy for estimating the winter oilseed rape LAI. The R^2^ of the validation set is 0.882, RMSE is 0.204, and MRE is 6.498%, representing an improvement of 36.3% in R2, and reductions of 37.0% and 40.0% in RMSE and MRE, respectively, compared to the traditional input (vegetation indices) model.

Finally, using the optimal estimation model (XGBoost_VIs + TFs + TTIs), we mapped the spatial distribution of winter oilseed rape LAI based on UAV imagery (Figure 7). The optimal model accurately predicted LAI for different treatments and effectively captured treatment differences.

## 4. Discussion

The LAI is used to provide information on the dynamic growth of crops and is an important parameter for monitoring crop growth [29]. Considering the applicability and sensitivity of UAV multispectral data in crop parameter monitoring [30], this study extracted UAV multispectral data for estimating the winter oilseed rape LAI.

Canopy spectral information has been widely used to estimate the crop LAI [30,31]. Different spectral information extracted by multispectral sensors has varying responses to the LAI. The LAI determines the absorption of photosynthetically active radiation (PAR) in the canopy to a certain extent [32]. Therefore, vegetation indices constructed with spectral bands are significantly correlated with the LAI. The study found that the re-normalized RDVI has the highest sensitivity to the LAI. This may be because RDVI combines the advantages of the NDVI and, through specific combinations of red and near-infrared light (NIR), reduces the influence of soil background and illumination changes. This makes RDVI more sensitive within the medium to high LAI range. In this range, the absorption of red light and the reflection of NIR are closely related to plant biomass and leaf structure. As the LAI increases, the canopy’s light absorption capacity enhances, especially in the red light band, leading to a decrease in red light reflection and an increase in NIR reflection. Finally, since RDVI uses the square root difference between red and NIR reflectance in its calculation, this difference more accurately reflects the changes in reflectance caused by variations in the LAI, thereby improving sensitivity to the LAI [33,34].

By extracting texture features from UAV images, the data types for estimating the winter oilseed rape LAI were increased, providing a potential technical means for evaluating crop parameters using UAV images. Additionally, to address the weak correlation between texture features and the LAI, a method for establishing texture indices was proposed, improving the performance of texture in estimating the LAI [35]. The results indicate that both two-dimensional and three-dimensional texture indices have higher correlation coefficients with the LAI compared to texture features alone. This is likely because constructing normalized texture indices can reduce the influence of soil background, solar angle, and sensor perspective [36]. The DTI can eliminate the interference of the same background in an image, while the ratio texture index can minimize the impact of topography and shadows on the image, amplifying ground features [37]. Overall, two-dimensional and three-dimensional texture indices can more comprehensively capture the spatial variability of surface vegetation by integrating multiple texture features. This multi-feature combination approach reduces the limitations of single texture features in information expression, enhancing the comprehensive performance of texture features at different scales and directions, thereby improving their correlation with the LAI. Additionally, compared to two-dimensional texture indices, three-dimensional texture indices further expand the dimensionality of texture information, enhancing the ability to describe complex surface structures. By extracting and combining texture features across multiple dimensions, these indices can make fuller use of information in images, thus showing higher accuracy in capturing the relationship between vegetation canopy structure and LAI [21].

The study found that integrating vegetation indices, texture features, and three-dimensional texture indices as model inputs yielded the highest LAI estimation accuracy. This is primarily because these three types of indicators cover different levels of information and can complement and enhance each other’s information expression during data fusion, thereby improving the model’s predictive ability. Vegetation indices are mainly based on spectral information, which can sensitively reflect the health status, density, and biomass of the vegetation canopy. However, they are often affected by saturation effects and may fail at high leaf area indices. In contrast, texture features analyze the spatial structure and texture information of images, capturing the spatial heterogeneity of the vegetation canopy and compensating for the deficiencies of vegetation indices in reacting to complex canopy structures [35]. The three-dimensional texture index integrates multiple texture features through the covariance matrix method, expanding the information dimensions and capturing richer texture information. This fusion of multi-dimensional features can more comprehensively characterize the complexity of the vegetation canopy, resulting in higher accuracy in LAI estimation [21]. Among the three modeling methods selected in this study, XGBoost outperformed SVM and PLSR in constructing LAI estimation models. This is likely because XGBoost, as an ensemble algorithm based on decision trees, can automatically handle non-linear relationships in data through tree splitting [18]. In contrast, SVM typically relies on kernel functions to handle non-linear features but may face increased complexity and computational costs in high-dimensional spaces [26]. On the other hand, PLSR is mainly based on linear regression and struggles to capture complex non-linear relationships [38]. Additionally, XGBoost employs regularization terms to prevent overfitting, particularly in high-dimensional data and small sample sizes, effectively avoiding the overfitting phenomenon and enhancing the model’s generalization ability. This allows XGBoost to more stably predict unknown data in multi-source remote sensing dataset modeling [39]. Consequently, XGBoost better adapts to the complex non-linear relationships between the LAI and various spectral parameters, improving model accuracy.

Despite the significant improvement in estimating the winter oilseed rape LAI by integrating vegetation indices, texture features, and three-dimensional texture indices, there are still some methodological limitations. Firstly, although three-dimensional texture indices expand the information dimensions, their combination formulas constructed through correlation matrix methods may not fully capture the high-order interactions between texture features, resulting in an incomplete description of the complex canopy structure. Secondly, the insufficient number of samples may limit the model’s generalization ability in high-dimensional feature spaces, especially at different growth stages or under varying environmental conditions, potentially limiting the model’s robustness and applicability [40]. This study has limitations, including a relatively small sample size and a focus on specific growth stages and environmental conditions, which may limit the model’s generalizability across diverse regions and rapeseed varieties. Additionally, key environmental variables (e.g., meteorological factors, soil properties) and complex texture feature interactions were not fully integrated, potentially reducing prediction accuracy. To further improve the accuracy of LAI estimation, future research should employ more refined feature selection methods to optimize the diversity and representativeness of model input variables. Additionally, increasing the sample size, especially samples from different growth stages and environmental conditions, can better capture the dynamic changes in the LAI, thereby enhancing the robustness and generalization ability of the model. Incorporating key environmental variables such as meteorological factors, soil characteristics, and canopy structure into the model can help build a more robust estimation framework, further enhancing the model’s applicability across different regions and crop varieties. By introducing higher-dimensional and multi-level texture feature combination strategies, particularly improvements in capturing potential interactions between texture features, will also provide strong support for improving the model’s prediction accuracy.

## 5. Conclusions

This study, based on field experiments and UAV multispectral data, constructed multi-source spectral parameters (vegetation indices, texture features, and texture indices) and employed SVM, PLSR, and XGBoost for LAI modeling. The results showed that the majority of vegetation indices had a significant correlation with the winter oilseed rape LAI (*p* < 0.05), with the highest correlation coefficient being the GNDVI at 0.715. Over half of the texture features had a significant correlation with the winter oilseed rape LAI (*p* < 0.05), with the highest correlation coefficient being the DIS value in band 5 at 0.603. All randomly combined texture indices had a significant correlation with the winter oilseed rape LAI (*p* < 0.05), and most three-dimensional texture indices had higher correlation coefficients than two-dimensional texture indices. Among them, NDTI was the most correlated two-dimensional texture index with the winter oilseed rape LAI, with a correlation coefficient of 0.676, located at the combination of CON5 and VAR3; NDTTI was the most correlated three-dimensional texture index with the winter oilseed rape LAI, with a correlation coefficient of 0.725, located at the combination of DIS5, VAR5, and VAR3. Combining vegetation indices with texture features and three-dimensional vegetation indices as input into the XGBoost model achieved the highest model accuracy for estimating the winter oilseed rape LAI, with an R^2^ of 0.882, RMSE of 0.204, and MRE of 6.498% on the validation set. Compared to the traditional input (vegetation indices) model, the R^2^ of the validation set increased by 36.3%, and the RMSE and MRE decreased by 37.0% and 40.0%, respectively. With the proliferation of UAV-based remote sensing platforms, this study provides a practical method for crop growth monitoring using UAV imagery.

## Figures and Tables

**Figure 1 plants-14-01245-f001:**
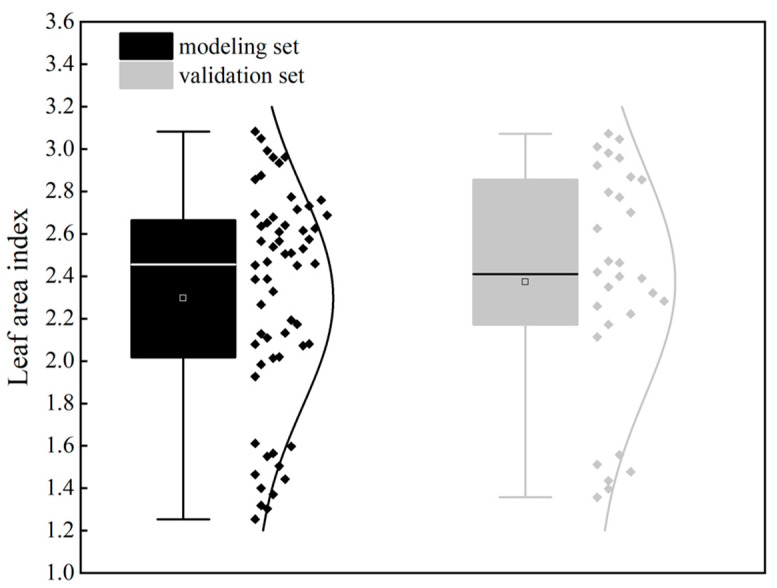
Statistical characteristics of winter oilseed rape leaf area index modeling set and validation set.

**Figure 2 plants-14-01245-f002:**
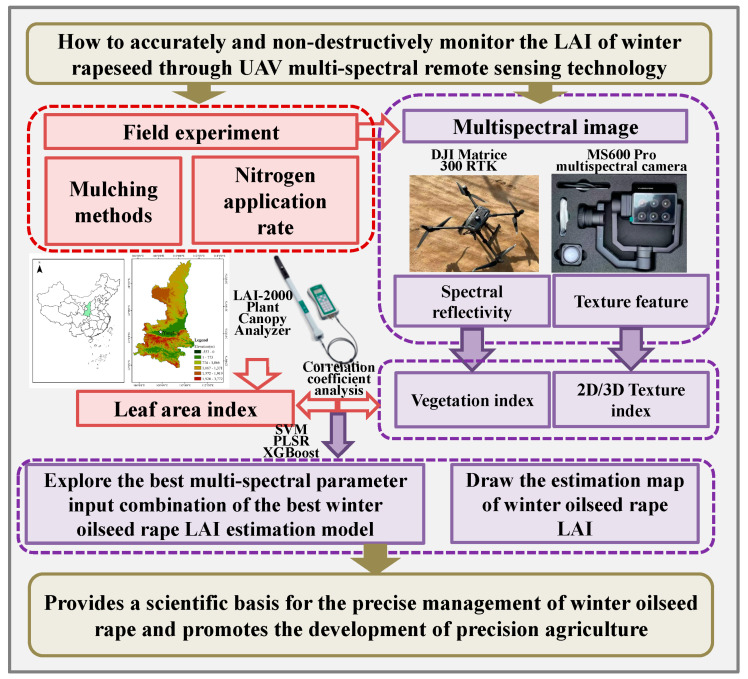
Flowchart of the proposed method.

**Figure 3 plants-14-01245-f003:**
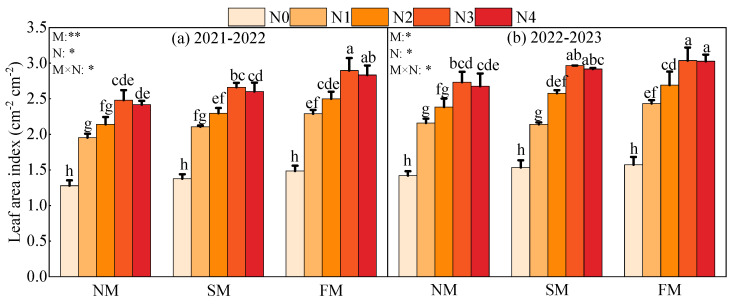
Effects of different mulching methods and nitrogen application rates on LAI of winter oilseed rape at bolting stage. The horizontal bars represent the standard deviation of the means (n = 3). Different alphabets indicate the significance within the same year at 5% level by LSD test. *, Significant at *p* < 0.05; **, Significant at *p* < 0.01. M represents the effect of mulching methods LAI of winter oilseed rape, N represents the effect of nitrogen application rate on LAI of winter oilseed rape, M × N represents the interaction between mulching methods and nitrogen application rate on LAI of winter oilseed rape.

**Figure 4 plants-14-01245-f004:**
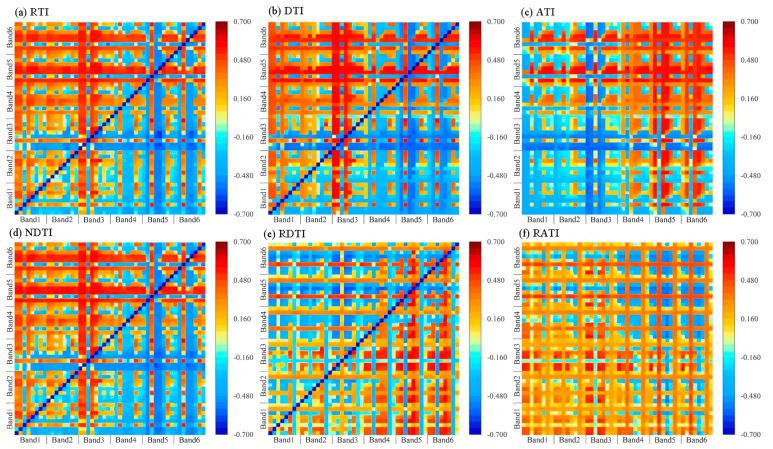
The correlation coefficient between leaf area index and two-dimensional texture index of winter oilseed rape (**a**) RTI, (**b**) DTI, (**c**) ATI, (**d**) NDTI, (**e**) RDTI, (**f**) RATI. Any point in the figure represents the correlation coefficient between the texture index and the leaf area index of winter rapeseed. The texture index is calculated by the two texture eigenvalues corresponding to the horizontal and vertical coordinates of the point.

**Figure 5 plants-14-01245-f005:**
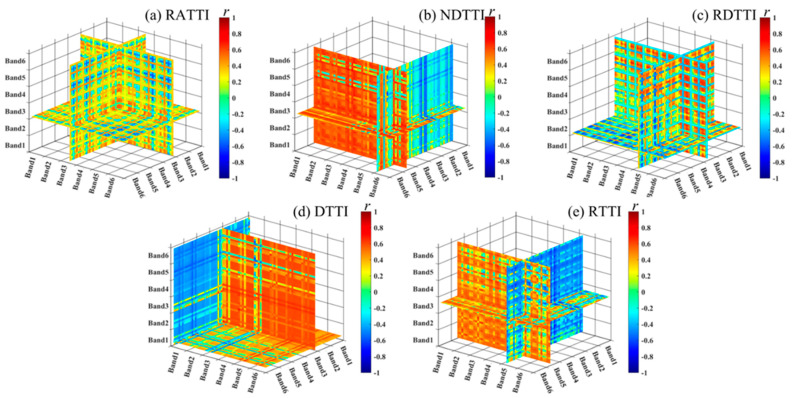
The correlation coefficients between leaf area index and three-dimensional texture index of winter rapeseed were (**a**) RATTI, (**b**) NDTTI, (**c**) RDTTI, (**d**) DTTI, (**e**) RTTI,. Any point in the figure represents the correlation coefficient between the texture index and the leaf area index of winter oilseed rape. The texture index is calculated by the three texture eigenvalues corresponding to the coordinates of the point.

**Figure 6 plants-14-01245-f006:**
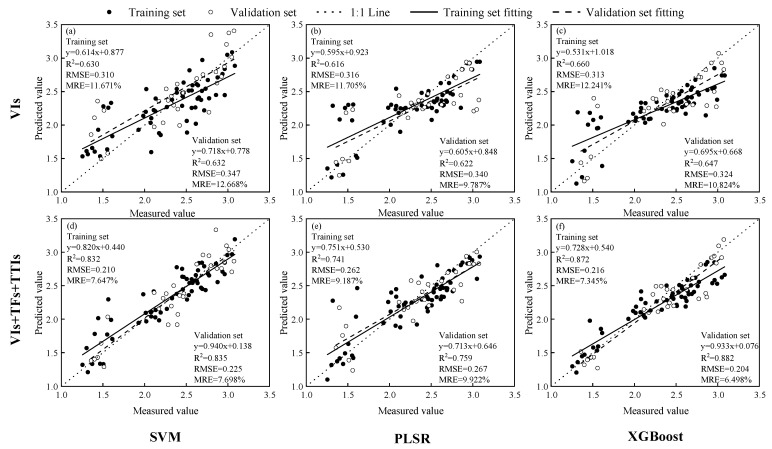
The input variables are the vegetation index (traditional input) and the multi-source remote sensing dataset (the best input in Table 8). The prediction results of the modeling set and the validation set of the winter oilseed rape leaf area index estimation model constructed by different machine learning models were compared. (**a**) LAI estimation model with vegetation index as input combined with SVM model, (**b**) LAI estimation model with vegetation index as input combined with PLSR model, (**c**) LAI estimation model with vegetation index as input combined with XGBoost model, (**d**) mixed input (vegetation index + texture feature + three-dimensional texture index) as input combined with SVM model, (**e**) mixed input (vegetation index + texture feature + three-dimensional texture index) as input combined with PLSR model, (**f**) mixed input (vegetation index + texture feature + three-dimensional texture index) as input combined with XGBoost model LAI estimation model.

**Figure 7 plants-14-01245-f007:**
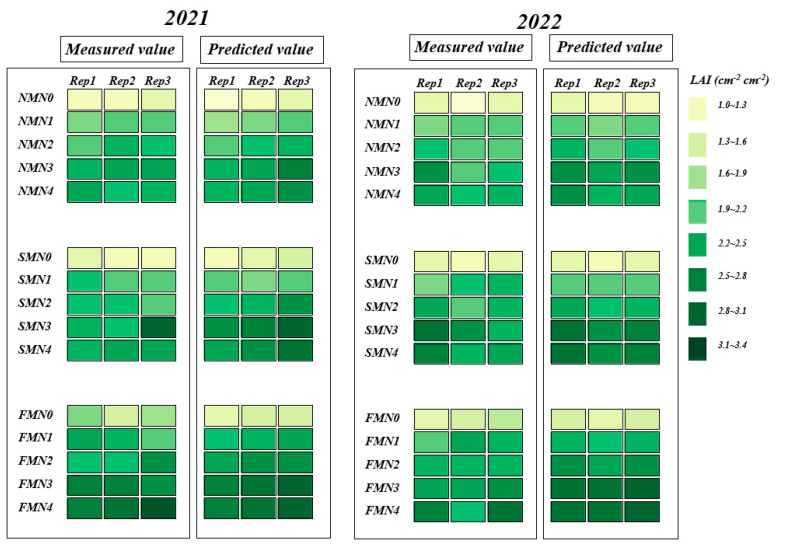
In the winter oilseed rape bolting stage, based on VIs + TFs + TTIs as input, the XGBoost model was used to estimate the estimated value of winter oilseed rape leaf area index and compared with the measured value.

**Table 1 plants-14-01245-t001:** Experimental design of split-plot treatments for winter oilseed rape. Factors: mulching methods (main plots) and nitrogen application rates (subplots).

Mulching Method	Nitrogen Application Rate/(kg·hm^−2^)	Treatment No.
No mulching	0	NMN0
	70	NMN1
	140	NMN2
	210	NMN3
	280	NMN4
Straw mulching	0	SMN0
	70	SMN1
	140	SMN2
	210	SMN3
	280	SMN4
Film mulching	0	FMN0
	70	FMN1
	140	FMN2
	210	FMN3
	280	FMN4

**Table 2 plants-14-01245-t002:** Spectral camera band information.

Spectral Waveband	Center Wavelength/nm	Bandwidth/nm	Whiteboard Reflectivity
Blue Light Band (B)	450	35	0.64
Green Light Band (G)	555	25	0.62
Red Light Band (R)	660	20	0.61
Red Edge Band 1 (RE1)	720	10	0.61
Red Edge Band 2 (RE2)	750	15	0.61
Near-Infrared Band (NIR)	840	35	0.57

**Table 5 plants-14-01245-t005:** Correlation analysis between vegetation index and leaf area index of winter oilseed rape.

Selection of Index	Correlation Coefficient
Soil-Adjusted Vegetation Index (SAVI)	0.676 *
Enhanced Vegetation Index (EVI)	0.647 *
Modified Simple Ratio (MSR)	0.610 *
Difference Vegetation Index (DVI)	0.607 *
Green Chlorophyll Index (CIgreen)	0.359 *
Renormalized Difference Vegetation Index (RDVI)	0.676 *
Transformed Vegetation Index (TVI)	0.644 *
Green Normalized Difference Vegetation Index (GNDVI)	0.715 *
Visible Difference Vegetation Index (VDVI)	0.676 *
Chlorophyll Vegetation Index (CVI)	0.162

Note: * Indicates significance at the *p* < 0.05 level.

**Table 6 plants-14-01245-t006:** Correlation analysis between texture features and leaf area index of winter oilseed rape.

Texture Feature	Correlation Coefficient					
	Band1	Band2	Band3	Band4	Band5	Band6
MEA	0.448 *	0.357 *	0.602 *	0.107	0.297 *	0.255 *
VAR	0.422 *	0.263 *	0.565 *	0.279 *	0.541 *	0.481 *
HOM	0.024	0.213 *	0.178	0.396 *	0.509 *	0.498 *
CON	0.393 *	0.200	0.545 *	0.312 *	0.561 *	0.502 *
DIS	0.286 *	0.034	0.456 *	0.409 *	0.603 *	0.567 *
ENT	0.167	0.277 *	0.102	0.338 *	0.379 *	0.375 *
SEM	0.252 *	0.303 *	0.222 *	0.332 *	0.353 *	0.352 *
COR	0.038	0.110	0.078	0.107	0.071	0.028

Note: * Indicates significance at the *p* < 0.05 level.

**Table 7 plants-14-01245-t007:** Correlation analysis between Texture index and leaf area index of winter oilseed rape.

Texture Index	The Maximum Value of Correlation Coefficient and Its Location Combination	
	Correlation Coefficient	Location Combination
RTI	0.635 *	DIS5, ENT6
DTI	0.660 *	VAR3, VAR5
ATI	0.620 *	DIS5, COR3
NDTI	0.676 *	CON5, VAR3
RDTI	0.665 *	DIS3, DIS5
RATI	0.602 *	CON3, MEA3
RATTI	0.576 *	MEA3, CON3, MEA3
NDTTI	0.725 *	DIS5, VAR5, VAR3
RDTTI	0.672 *	DIS3, VAR5, MEA2
DTTI	0.701 *	MEA3, MEA1, VAR1
RTTI	0.650 *	DIS5, VAR5, DIS3

Note: * Indicates significance at the *p* < 0.05 level.

**Table 8 plants-14-01245-t008:** Comparison of accuracy test results of the estimation model validation set.

Input Combination	Validation Set Model Evaluation Index	SVM	PLSR	XGBoost
VIs	R^2^	0.632	0.622	0.647
	RMSE	0.347	0.340	0.324
	MRE (%)	12.668	9.787	10.824
TFs	R^2^	0.495	0.464	0.581
	RMSE	0.343	0.356	0.317
	MRE (%)	13.558	14.323	12.882
TIs	R^2^	0.673	0.635	0.687
	RMSE	0.256	0.287	0.305
	MRE (%)	8.794	10.989	10.804
TTIs	R^2^	0.740	0.715	0.763
	RMSE	0.279	0.296	0.211
	MRE (%)	11.321	11.901	7.157
VIs + TFs	R^2^	0.714	0.695	0.729
	RMSE	0.349	0.285	0.306
	MRE (%)	11.573	10.726	11.467
VIs + TIs	R^2^	0.752	0.744	0.768
	RMSE	0.228	0.237	0.201
	MRE (%)	7.264	7.349	7.035
VIs + TTIs	R^2^	0.812	0.751	0.824
	RMSE	0.265	0.276	0.253
	MRE (%)	8.639	8.943	8.157
TFs + TIs	R^2^	0.644	0.637	0.717
	RMSE	0.357	0.352	0.301
	MRE (%)	11.578	11.641	11.429
TFs + TTIs	R^2^	0.783	0.749	0.797
	RMSE	0.359	0.364	0.213
	MRE (%)	9.461	9.729	8.052
TIs + TTIs	R^2^	0.672	0.657	0.756
	RMSE	0.330	0.333	0.285
	MRE (%)	11.155	11.028	10.784
VIs + TFs + TIs	R^2^	0.796	0.735	0.845
	RMSE	0.242	0.352	0.207
	MRE (%)	8.129	9.438	5.393
VIs + TFs + TTIs	R^2^	0.835	0.759	0.882
	RMSE	0.225	0.267	0.204
	MRE (%)	7.698	9.922	6.498
VIs + TIs + TTIs	R^2^	0.733	0.714	0.781
	RMSE	0.277	0.362	0.210
	MRE (%)	9.657	11.973	4.864
TFs + TIs + TTIs	R^2^	0.720	0.704	0.763
	RMSE	0.290	0.378	0.222
	MRE (%)	9.991	12.387	5.132
VIs + TFs + TIs + TTIs	R^2^	0.793	0.784	0.816
	RMSE	0.283	0.386	0.351
	MRE (%)	9.351	9.724	8.573

## Data Availability

The original contributions presented in this study are included in the article. Further inquiries can be directed to the corresponding authors.

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
