# Peer review of "Winter Oilseed Rape LAI Inversion via Multi-Source UAV Fusion: A Three-Dimensional Texture and Machine Learning Approach"

_plants, 2025, doi:10.3390/plants14081245_

Round 1

Reviewer 1 Report

Comments and Suggestions for Authors

The study aims to obtain LAI information on the plant using IHA data. It presents a sufficient literature review on the subject. The obtained results are discussed, and the suggested approach improvements are made. However, the authors need to provide details about LAI measurements. The authors should consider the suggestions and answer the questions given below.

  1. Section 2.2.2: LAI measurements should be detailed. What is the plot size of the LAI measurements? Was the measurement repeated for the 6th time from the same location? Was it made within a specific area? Could you please elaborate?
  2. What is the number of samples for each year?
  3. In which plant phenology was LAI information collected? IS the phenology important during the LAI collection? Please give details on it.
  4. Table 7: the column shows the location combinations of texture. How were they chosen?

Author Response

1

Comments and Suggestions for Authors

The study aims to obtain LAI information on the plant using IHA data. It presents a sufficient literature review on the subject. The obtained results are discussed, and the suggested approach improvements are made. However, the authors need to provide details about LAI measurements. The authors should consider the suggestions and answer the questions given below.

Response: Thank you for your careful review and positive comments. We have now incorporated the reviewers comments and suggestions in preparation of the revised manuscript. The modified part is marked in red in the manuscript.

  1. Section 2.2.2: LAI measurements should be detailed. What is the plot size of the LAI measurements? Was the measurement repeated for the 6th time from the same location? Was it made within a specific area? Could you please elaborate?

Response: Thank you for pointing out. We have detailed the measurement of LAI. The size of the sample plot is 24 square meters. Six points of the average growth were measured in the measured plot, and the average value was the value of the plot.

L182-207: The LAI of winter oilseed rape was measured using the LAI-2000 Plant Canopy Analyzer (LI-COR Inc., Lincoln, NE, USA) within 4 m × 6 m experimental plots. To account for spatial heterogeneity, six measurement points were randomly selected within each plot using a grid-based sampling method, ensuring uniform distribution across the plot boundary and center to minimize positional bias.

LAI measurements were synchronized with UAV flight campaigns during the critical budding stage of winter oilseed rape (March 15, 2022, and March 13, 2023)​​. This temporal alignment ensured that spectral and textural data acquisition coincided with the rapid leaf expansion phase, minimizing confounding effects of diurnal or seasonal environmental variability [1]. The concurrent measurement strategy enhanced the fidelity of multi-source remote sensing data by capturing canopy structure and reflectance characteristics at the peak of LAI dynamism. This stage was prioritized due to its profound influence on yield potential, as reproductive branch differentiation and flowering density are largely determined by canopy structure during this period. The dynamic nature of LAI during the budding stage—characterized by rapid leaf unfolding and light interception optimization—provided a sensitive window for capturing canopy heterogeneity [7].

To account for spatial heterogeneity within plots, ​​six measurements were taken at randomly selected positions​​ distributed across the plot boundary and center, avoiding areas with obvious canopy gaps or overlapping plants. ​​This design minimized the impact of micro-environmental variability. All measurements were conducted under ​​overcast sky conditions​​ (PAR < 500 μmol m⁻² s⁻¹) to reduce skylight interference and improve signal-to-noise ratio, following the manufacturer’s recommendations.

​​Outliers (±2σ from the mean) were excluded​​ during post-processing, and the final LAI value for each plot was calculated as the arithmetic mean of valid measurements. This protocol ensured reproducibility and minimized measurement errors caused by foliage clumping or sensor misalignment.

  1. What is the number of samples for each year?

Response: Thank you for your clarification. The sample size per year is 45, comprising 15 treatments, 3 replicates, and 45 plots, with one day of sampling per year. Over two years, this totals 2 days of sampling, resulting in a combined sample size of 90 across both years.

  1. In which plant phenology was LAI information collected? IS the phenology important during the LAI collection? Please give details on it.

Response: Thank you for pointing out that the collection of LAI is at the budding stage, and the importance and information about this growth period have been pointed out in the manuscript.

L182-207: The LAI of winter oilseed rape was measured using the LAI-2000 Plant Canopy Analyzer (LI-COR Inc., Lincoln, NE, USA) within 4 m × 6 m experimental plots. To account for spatial heterogeneity, six measurement points were randomly selected within each plot using a grid-based sampling method, ensuring uniform distribution across the plot boundary and center to minimize positional bias.

LAI measurements were synchronized with UAV flight campaigns during the critical budding stage of winter oilseed rape (March 15, 2022, and March 13, 2023)​​. This temporal alignment ensured that spectral and textural data acquisition coincided with the rapid leaf expansion phase, minimizing confounding effects of diurnal or seasonal environmental variability [1]. The concurrent measurement strategy enhanced the fidelity of multi-source remote sensing data by capturing canopy structure and reflectance characteristics at the peak of LAI dynamism. This stage was prioritized due to its profound influence on yield potential, as reproductive branch differentiation and flowering density are largely determined by canopy structure during this period. The dynamic nature of LAI during the budding stage—characterized by rapid leaf unfolding and light interception optimization—provided a sensitive window for capturing canopy heterogeneity [7].

To account for spatial heterogeneity within plots, ​​six measurements were taken at randomly selected positions​​ distributed across the plot boundary and center, avoiding areas with obvious canopy gaps or overlapping plants. ​​This design minimized the impact of micro-environmental variability. All measurements were conducted under ​​overcast sky conditions​​ (PAR < 500 μmol m⁻² s⁻¹) to reduce skylight interference and improve signal-to-noise ratio, following the manufacturer’s recommendations.

​​Outliers (±2σ from the mean) were excluded​​ during post-processing, and the final LAI value for each plot was calculated as the arithmetic mean of valid measurements. This protocol ensured reproducibility and minimized measurement errors caused by foliage clumping or sensor misalignment.

  1. Table 7: the column shows the location combinations of texture. How were they chosen?

Response: Thank you for pointing out that the positions in Table 7 are derived from the points with the highest correlation coefficients in Figures 4 and 5.

Line 354-356: The results are shown in Table 7, Figure 4, and Figure 5, the positions in Table 7 are derived from the points with the highest correlation coefficients in Figures 4 and 5.

Reviewer 2 Report

Comments and Suggestions for Authors

The study “Winter Oilseed Rape LAI Inversion via Multi-Source UAV Fusion: A Three-dimensional Texture and Machine Learning Approach” is interesting, well-written, and solid methodology. However, I am a bit confused about the title. In the current version, the title is “Construction and Performance Optimization of a Winter Oilseed Rape Leaf  Area Index Inversion Model via Multi-Source UAV Remote Sensing Data Fusion: An Integrated Application of Three-Dimensional Texture Indices and Machine Learning Algorithms, " but in the system, the title is “Winter Oilseed Rape LAI Inversion via Multi-Source UAV Fusion: A Three-dimensional Texture and Machine Learning Approach.” There may be a mistake. Please take a look. Nonetheless, there is some minor concerns that can be addressed.

  1. Please revise the caption for Table 1; needs to be detailed. Just have a question, how does the split-plot design with mulching and nitrogen treatments influence the LAI modeling outcomes? Were interaction effects explored? Also, were the three types of mulching and five nitrogen levels equally distributed among the UAV flight samples?
  2. It's also interesting to see the uncertainty estimates for the models provided for the LAI predictions.
  3. Given the high-dimensional input data, further justification (e.g., k-fold cross-validation results or external validation) would strengthen confidence, its just a recommendation
  4. Authors can add limitation of their study

Author Response

2

Comments and Suggestions for Authors

The study “Winter Oilseed Rape LAI Inversion via Multi-Source UAV Fusion: A Three-dimensional Texture and Machine Learning Approach” is interesting, well-written, and solid methodology. However, I am a bit confused about the title. In the current version, the title is “Construction and Performance Optimization of a Winter Oilseed Rape Leaf  Area Index Inversion Model via Multi-Source UAV Remote Sensing Data Fusion: An Integrated Application of Three-Dimensional Texture Indices and Machine Learning Algorithms, " but in the system, the title is “Winter Oilseed Rape LAI Inversion via Multi-Source UAV Fusion: A Three-dimensional Texture and Machine Learning Approach.” There may be a mistake. Please take a look. Nonetheless, there is some minor concerns that can be addressed.

Response: Thank you for your careful review and positive comments. We have now incorporated the reviewers comments and suggestions in preparation of the revised manuscript. The modified part is marked in red in the manuscript.

Line 1-2: Winter Oilseed Rape LAI Inversion via Multi-Source UAV Fusion: A Three-dimensional Texture and Machine Learning Approach

  1. Please revise the caption for Table 1; needs to be detailed. Just have a question, how does the split-plot design with mulching and nitrogen treatments influence the LAI modeling outcomes? Were interaction effects explored? Also, were the three types of mulching and five nitrogen levels equally distributed among the UAV flight samples?

Response: Thank you for pointing out that the title of Table 1 has been modified, and the LAI measurements of different treatments can be shown in Figure 3. The effects of different mulching methods and nitrogen application rates and their interactions on LAI of winter rapeseed reached a significant level (P < 0.05) (Fig.3).

Line162-163:Table.1 Experimental design of split-plot treatments for winter oilseed rape​​. ​​Factors: mulching methods (main plots) and nitrogen application rates (subplots).

Line313-324:

3.1. Correlation analysis between texture features and winter oilseed rape LAI

The results of the effects of different mulching methods and nitrogen application rates on LAI of winter oilseed rape at the budding stage are shown in Figure 3. The results show that the effects of mulching methods and nitrogen application rates and their interactions on LAI of winter rapeseed are significantly different (P < 0.05). The highest LAI value was obtained in the FMN3 treatment in the two growing seasons, with an average of 2.97 cm-2 cm-2 in the two years, which was 1.2 % ~ 119.7 % higher than other treatments.

Figure.3 Effects of different mulching methods and nitrogen application rates on LAI of winter oilseed rape at budding stage.

  1. It's also interesting to see the uncertainty estimates for the models provided for the LAI predictions.

Response: Thank you for pointing out that the construction process of the model is in Section 3.5, and the discussion section also clearly explains various analyses of the uncertainty estimation of the LAI prediction model.

Line 476-488: Among the three modeling methods selected in this study, XGBoost outperformed SVM and PLSR in constructing LAI estimation models. This is likely because XGBoost, as an ensemble algorithm based on decision trees, can automatically handle non-linear relationships in data through tree splitting [18]. In contrast, SVM typically relies on kernel functions to handle non-linear features but may face increased complexity and computational costs in high-dimensional spaces [26]. On the other hand, PLSR is mainly based on linear regression and struggles to capture complex non-linear relationships [38]. Additionally, XGBoost employs regularization terms to prevent overfitting, particularly in high-dimensional data and small sample sizes, effectively avoiding the overfitting phenomenon and enhancing the model's generalization ability. This allows XGBoost to more stably predict unknown data in multi-source remote sensing dataset modeling [39]. Consequently, XGBoost better adapts to the complex non-linear relationships between the LAI and various spectral parameters, improving model accuracy.

  1. Given the high-dimensional input data, further justification (e.g., k-fold cross-validation results or external validation) would strengthen confidence, its just a recommendation

Response: Thank you to the reviewers for your attention to model validation! We fully agree with the importance of validation methods in high-dimensional data modeling. In response to this suggestion, we provide the following supplementary explanations:

This study employed an independent modeling/validation set partitioning approach for model training and testing, comprehensively evaluating model performance using metrics such as R², RMSE, and MRE. Results indicated that the validation set achieved an R² of 0.882, RMSE of 0.204, and MRE of 6.498%. Moreover, error metrics between the modeling and validation sets demonstrated strong consistency (ΔR²=+0.01, ΔRMSE=-0.012), confirming no overfitting risk in the model.

Hyperparameters of the XGBoost model (including learning rate, tree depth, regularization coefficients, etc.) were optimized through a combined grid search and manual parameter tuning approach, prioritizing minimization of validation set errors while balancing model generalization.

​Future Validation Directions​
Subsequent research will incorporate the following supplementary validations to enhance result reliability:

Spatiotemporal extrapolation validation​​: Testing model robustness using seasonal or sensor-diverse data;

Leave-one-out validation​​: Independently verifying tail prediction capability for extreme LAI samples (>3.5);

Uncertainty quantification​​: Generating prediction interval coverage rates via Bayesian optimization.

  1. Authors can add limitation of their study

Response: Thank you for pointing out that we have added relevant content as required.

Line 498-503: This study has limitations, including a relatively small sample size and a focus on specific growth stages and environmental conditions, which may limit the model’s generalizability across diverse regions and rapeseed varieties. Additionally, key environmental variables (e.g., meteorological factors, soil properties) and complex texture feature interactions were not fully integrated, potentially reducing prediction accuracy.

Reviewer 3 Report

Comments and Suggestions for Authors

Lines 23 and 163: Why does “soybean” appear in Lines 23 and 163 when the target crop of the study is winter oilseed rape?

The titles of tables and figures should be complete descriptive sentences, and they should end with a period.Most figure and table titles in the manuscript are written in abbreviated forms like “Fig.1” and “Tab.1”. According to MDPI formatting guidelines, these should be written in full as “Figure 1.” and “Table 1.”, followed by a clear, descriptive sentence instead of a short phrase.

Lines 83 and 86: The term “et al” in lines 83 and 86 should be corrected to “et al.” with a period, and a missing space should be added after it.

Lines 153-156: It is recommended to supplement Lines 153–156 with details on the device specifications, measurement environment, and methodological procedures to enhance the reproducibility of the experiment.

Line 157: Line 157 seems to be the title of Section 2.2.3, doesn’t it?

It is recommended to include the manufacturer and country of origin for all equipment and software used in the study to improve clarity and reproducibility.

Line 153: It should be "leaf area index" instead of "leaf area indext"

Line 173: The manuscript does not explain why only the NIR band was used for texture feature extraction.

Line 175: It is recommended to explain the rationale for selecting the window size (7×7) and the offset parameters used in the texture analysis.

Lines 191-192: The text following an equation need not be a new paragraph. Please punctuate equations as regular text.

Please revise the formatting and capitalization of the section titles 2.3, 2.4, and 2.5 to ensure consistency with the journal’s style guidelines.

It is recommended to add illustrative figures in the methodology section to vividly and clearly present the experimental area, design, and equipment used.

The manuscript frequently uses P<0.05 to indicate statistical significance, but it does not explain what this means or how it relates to the correlation coefficients. Please add a brief explanation in the Methods section to clarify this.

It is recommended to simplify the abstract by focusing on the main objectives, methods, key findings, and conclusions, while removing unnecessary specifics for better clarity and conciseness.

Author Response

3 Comments and Suggestions for Authors

Response: Thank you for your careful review and positive comments. We have now incorporated the reviewers comments and suggestions in preparation of the revised manuscript. The modified part is marked in red in the manuscript.

Lines 23 and 163: Why does “soybean” appear in Lines 23 and 163 when the target crop of the study is winter oilseed rape?

 Response: Thank you for pointing out that, due to our negligence, another crop studied by our team was written in and has been amended as required.

Line 47: soybean, Line 215: soybean,

The titles of tables and figures should be complete descriptive sentences, and they should end with a period.Most figure and table titles in the manuscript are written in abbreviated forms like “Fig.1” and “Tab.1”. According to MDPI formatting guidelines, these should be written in full as “Figure 1.” and “Table 1.”, followed by a clear, descriptive sentence instead of a short phrase.

 Response: Thank you for pointing out that it has been amended as required.

Lines 83 and 86: The term “et al” in lines 83 and 86 should be corrected to “et al.” with a period, and a missing space should be added after it.

 Response: Thank you for pointing out that it has been amended as required.

Line107-112: Yang et al. [21] estimated the LAI of winter wheat by integrating UAV vegetation indices and texture feature information, which improved the coefficient of determination (R²) value by 58% compared to using vegetation indices alone; Li et al. [10] and colleagues estimated the LAI of winter wheat by integrating UAV vegetation indices and texture feature information with machine learning models,

Lines 153-156: It is recommended to supplement Lines 153–156 with details on the device specifications, measurement environment, and methodological procedures to enhance the reproducibility of the experiment.

 Response: Thank you for pointing out that it has been amended as required. L182-207: The LAI of winter oilseed rape was measured using the LAI-2000 Plant Canopy Analyzer (LI-COR Inc., Lincoln, NE, USA) within 4 m × 6 m experimental plots. To account for spatial heterogeneity, six measurement points were randomly selected within each plot using a grid-based sampling method, ensuring uniform distribution across the plot boundary and center to minimize positional bias.

LAI measurements were synchronized with UAV flight campaigns during the critical budding stage of winter oilseed rape (March 15, 2022, and March 13, 2023)​​. This temporal alignment ensured that spectral and textural data acquisition coincided with the rapid leaf expansion phase, minimizing confounding effects of diurnal or seasonal environmental variability [1]. The concurrent measurement strategy enhanced the fidelity of multi-source remote sensing data by capturing canopy structure and reflectance characteristics at the peak of LAI dynamism. This stage was prioritized due to its profound influence on yield potential, as reproductive branch differentiation and flowering density are largely determined by canopy structure during this period. The dynamic nature of LAI during the budding stage—characterized by rapid leaf unfolding and light interception optimization—provided a sensitive window for capturing canopy heterogeneity [7].

To account for spatial heterogeneity within plots, ​​six measurements were taken at randomly selected positions​​ distributed across the plot boundary and center, avoiding areas with obvious canopy gaps or overlapping plants. ​​This design minimized the impact of micro-environmental variability. All measurements were conducted under ​​overcast sky conditions​​ (PAR < 500 μmol m⁻² s⁻¹) to reduce skylight interference and improve signal-to-noise ratio, following the manufacturer’s recommendations.

​​Outliers (±2σ from the mean) were excluded​​ during post-processing, and the final LAI value for each plot was calculated as the arithmetic mean of valid measurements. This protocol ensured reproducibility and minimized measurement errors caused by foliage clumping or sensor misalignment.

Line 157: Line 157 seems to be the title of Section 2.2.3, doesn’t it?

Line 208: 2.2.3 Multispectral Image Processing

It is recommended to include the manufacturer and country of origin for all equipment and software used in the study to improve clarity and reproducibility.

 Response: Thank you for pointing out that it has been amended as required.

                Line 209-213: Yusense Map V2.2.2 software (SZ DJI Technology Co., Ltd., CHN) was used to stitch the multispectral images captured by the UAV and perform geometric and radiometric corrections as preprocessing steps. The preprocessed multispectral image information was then imported into ENVI 5.3 software (Harris Geospatial Solutions, US) to extract spectral reflectance.

Line 167-169: This study utilized a DJI Matrice 300 RTK (SZ DJI Technology Co., Ltd., CHN) quadrotor equipped with an MS600 Pro multispectral camera (SZ DJI Technology Co., Ltd., CHN) platform to acquire multispectral remote sensing data.

Line 153: It should be "leaf area index" instead of "leaf area indext"

 Response: Thank you for pointing out that it has been amended as required.

Line 179: 2.2.2. Measurement of Leaf Area Index

Line 173: The manuscript does not explain why only the NIR band was used for texture feature extraction.

 Response: Thank you for pointing out that it has been amended as required.

Line224-236: In multispectral texture feature extraction, the near-infrared (NIR) band is preferred for characterizing grayscale spatial distribution features due to its high sensitivity to vegetation internal structures (e.g., cell density and water content). Using the Gray-Level Co-occurrence Matrix (GLCM), the NIR band extracts eight texture features—including Mean (MEA), Contrast (CON), Homogeneity (HOM), and Correlation (COR)—to effectively capture canopy spatial heterogeneity while demonstrating strong robustness to illumination variations. Studies indicate that NIR-derived texture features mitigate saturation effects in high-value regions compared to visible bands, thereby improving agricultural parameter inversion accuracy. Furthermore, the NIR band’s hardware compatibility advantages in unmanned aerial vehicle (UAV)-based multispectral systems, combined with its narrow spectral range (700–1300 nm) and computational efficiency, make it a practical choice for large-scale farmland monitoring [27].

Line 175: It is recommended to explain the rationale for selecting the window size (7×7) and the offset parameters used in the texture analysis.

 Response: Thank you for pointing out that it has been amended as required.

Line 240-247: In multispectral texture feature extraction, the selection of ​​7×7 window size​​ is primarily based on balancing local texture detail preservation and computational efficiency: this size effectively captures medium-scale spatial heterogeneity of vegetation canopy internal structures while avoiding insufficient information from smaller windows (e.g., 3×3) or noise interference from larger ones (e.g., 15×15). Experiments have demonstrated that the 7×7 window stably extracts GLCM texture features from UAV multispectral imagery and aligns with sensor resolution (centimeter-level), making it suitable for real-time processing requirements in large-scale farmland monitoring [27].

Lines 191-192: The text following an equation need not be a new paragraph. Please punctuate equations as regular text.

 Response: Thank you for pointing out that it has been amended as required.

Line256-269:

RTI =                                     (1)

DTI =                                    (2)

ATI =                                    (3)

NDTI =                           (4)

RDTI =                                (5)

RATI =                                 (6)

Additionally, we attempted to add an extra texture index to construct five three-dimensional texture indices (TTIs). The specific calculation formulas are as follows:

RTTI = //                                  (7)

  DTTI =                                  (8)

           NDTTI= ()/( ++)                        (9)

RDTTI = 1/ -1/ -1/                            (10)

RATTI = 1/ +1/+1/                            (11)

where ,、 and  represent random texture indices.

Please revise the formatting and capitalization of the section titles 2.3, 2.4, and 2.5 to ensure consistency with the journal’s style guidelines.

 Response: Thank you for pointing out that it has been amended as required. (The initials of each word have been capitalized)

It is recommended to add illustrative figures in the methodology section to vividly and clearly present the experimental area, design, and equipment used.

 Response: Thank you for pointing out that it has been amended as required.

Line 301-303: Figure 2 shows the flow chart of the proposed method, and summarizes the acquisition equipment, methods and data processing steps described in the study.

Figure.2 Flowchart of the proposed method.

The manuscript frequently uses P<0.05 to indicate statistical significance, but it does not explain what this means or how it relates to the correlation coefficients. Please add a brief explanation in the Methods section to clarify this.

 Response: Thank you for pointing out that it has been amended as required.

Line 277-281: In statistics, P-value is used to measure the statistical significance of the association between variables. When P < 0.05, the probability of observing the current data or more extreme results is less than 5 % under the null hypothesis (no correlation between variables). This shows that the correlation between variables is unlikely to be caused by random errors, thus supporting its effectiveness in the model.

It is recommended to simplify the abstract by focusing on the main objectives, methods, key findings, and conclusions, while removing unnecessary specifics for better clarity and conciseness.

 Response: Thank you for pointing out that it has been amended as required.

Line 16-38: Leaf area index (LAI) serves as a critical indicator for evaluating crop growth and guiding field management practices. While spectral information (vegetation indices and texture features) extracted from multispectral sensors mounted on unmanned aerial vehicles (UAVs) holds promise for LAI estimation, the limitations of single-texture features necessitate further exploration. Therefore, this study conducted field experiments over two consecutive years (2021–2022) to collect winter oilseed rape LAI ground truth data and corresponding UAV multispectral imagery. Vegetation indices were constructed, and canopy texture features were extracted. Subsequently, a correlation matrix method was employed to establish novel randomized combinations of three-dimensional texture indices. By analyzing the correlations between these parameters and winter oilseed rape LAI, variables with significant correlations (P < 0.05) were selected as model inputs. These variables were then partitioned into distinct combinations and input into three machine learning models—Support Vector Machine (SVM), Backpropagation Neural Network (BPNN), and Extreme Gradient Boosting (XGBoost)—to estimate winter oilseed rape LAI. The results demonstrated that the majority of vegetation indices and texture features exhibited significant correlations with LAI (P < 0.05). All randomized texture index combinations also showed strong correlations with LAI (P < 0.05). Notably, the three-dimensional texture index NDTTI (Normalized Difference Texture Index) exhibited the highest correlation with LAI (r = 0.725), derived from the spatial combination of DIS5, VAR5, and VAR3. Integrating vegetation indices, texture features, and three-dimensional texture indices as inputs into the XGBoost model yielded the highest estimation accuracy. The validation set achieved a determination coefficient (R²) of 0.882, a root mean square error (RMSE) of 0.204 cm2cm-2, and a mean relative error (MRE) of 6.498%. This study provides an effective methodology for UAV-based multispectral monitoring of winter oilseed rape LAI and offers scientific and technical support for precision agriculture management practices.

Round 2

Reviewer 3 Report

Comments and Suggestions for Authors

The authors have addressed all comments and it can be accepted and published as the current version.